# Protist–Lactic Acid Bacteria Co-Culture as a Strategy to Bioaccumulate Polyunsaturated Fatty Acids in the Protist *Aurantiochytrium* sp. T66

**DOI:** 10.3390/md21030142

**Published:** 2023-02-23

**Authors:** Luis Marileo, Jacqueline Acuña, Joaquin Rilling, Paola Díaz, Antonio Luca Langellotti, Giovanni Luca Russo, Patricio Javier Barra, Patricio Dantagnan, Sharon Viscardi

**Affiliations:** 1Programa de Doctorado en Ciencias Agropecuarias, Facultad de Recursos Naturales, Universidad Católica de Temuco, Rudecindo Ortega 02950, Temuco 4780000, Chile; 2Biotechnology of Functional Foods Laboratory, Camino Sanquilco, Parcela 18, Padre Las Casas 4850827, Chile; 3Laboratorio de Ecología Microbiana Aplicada (EMALAB), Departamento de Ciencias Químicas y Recursos Naturales, Universidad de La Frontera, Temuco 4811230, Chile; 4Center of Plant, Soil Interaction and Natural Resources Biotechnology, Scientific and Technological Bioresource Nucleus (BIOREN), Universidad de La Frontera, Temuco 4811230, Chile; 5Departamento de Ciencia Agropecuarias y Acuícolas, Facultad de Recursos Naturales, Universidad Católica de Temuco, Temuco 4780694, Chile; 6Núcleo de Investigación en Producción Alimentaria, Universidad Católica de Temuco, Rudecindo Ortega 02950, Temuco 4780694, Chile; 7Center for Innovation and Developmentin Food Industry CAISIAL, University of Naples Federico II, Via Università, 133 Portici, Italy; 8Scientific and Technological Bioresource Nucleus (BIOREN), Universidad de La Frontera, Temuco 4780000, Chile; 9Laboratorio de Investigación en Salud de Precisión, Departamento de Procesos Diagnóstico y Evaluación, Facultad de Ciencias de la Salud, Universidad Católica de Temuco, Manuel Montt 56, Campus San Francisco, Temuco 4791086, Chile

**Keywords:** *Aurantiochytrium*, PUFA, LAB, co-culture, protist

## Abstract

Thraustochytrids are aquatic unicellular protists organisms that represent an important reservoir of a wide range of bioactive compounds, such as essential polyunsaturated fatty acids (PUFAs) such as arachidonic acid (ARA), docosahexaenoic acid (DHA), eicosapentaenoic acid (EPA), which are involved in the regulation of the immune system. In this study, we explore the use of co-cultures of *Aurantiochytrium* sp. and bacteria as a biotechnological tool capable of stimulating PUFA bioaccumulation. In particular, the co-culture of lactic acid bacteria and the protist *Aurantiochytrium* sp. T66 induce PUFA bioaccumulation, and the lipid profile was evaluated in cultures at different inoculation times, with two different strains of lactic acid bacteria capable of producing the tryptophan dependent auxins, and one strain of *Azospirillum* sp., as a reference for auxin production. Our results showed that the *Lentilactobacillus kefiri* K6.10 strain inoculated at 72 h gives the best PUFA content (30.89 mg g^−1^ biomass) measured at 144 h of culture, three times higher than the control (8.87 mg g^−1^ biomass). Co-culture can lead to the generation of complex biomasses with higher added value for developing aquafeed supplements.

## 1. Introduction

Polyunsaturated fatty acids (PUFAs) are fatty acids containing two or more double bonds between carbon atoms. The most important among them are the essential PUFAs, such as arachidonic acid (ARA; 20:4 *n*-6), docosahexaenoic acid (DHA, 22:6 *n*-3) and eicosapentaenoic acid (EPA) are essential components of cell membrane phospholipids of animals, including mammals and fish. These fatty acids play a major role in growth [1,2,3], lipid metabolism [2,4], health and disease resistance of fish [3,5]. Many carnivorous fish, including salmonids, are unable to synthesize long-chain PUFAs in sufficient amounts. Therefore, fish production is strongly dependent on dietary PUFA supplementation to meet their metabolic and physiological demands [6]. Traditionally, industrial fish production is based on dietary supplementation with fishmeal and fish oil for the formulation of feedstuffs. Nevertheless, this practice is being questioned due to their high cost and profound environmental impact [7]. Accordingly, it is essential to generate sustainable strategies that allow improving fish nutrition and production while minimizing costs and environmental impact of this growing industry [8,9,10].

During the last few years, the use of oil plants emerged as a strategy for the partial replacement of fishmeal and fish oil for reducing the cost of feed formulation. Nevertheless, the addition of plant biomass is limited for three main reasons: high levels of antinutritional factors, an imbalance of essential amino acids, and low levels of long-chain PUFAs [11]. Faced with this scenario, the incorporation of functional feed additives from unicellular marine microorganisms, such as protists, yeasts and bacteria, have been proposed as a promising biotechnological alternative of proteins and lipids for fish farming [12]. Marine microorganisms can provide a wide range of biocompounds, such as amino acids, antioxidants, carotenoids, vitamins, minerals, pigments (phycoerythrin and phycocyanin) and essential lipids such as PUFAs [13,14,15]. Thraustochytrids protists are a diverse group of unicellular heterotrophic eukaryotic microorganisms, of great ecological importance in both marine and freshwater environments. They have been noted for their high growth rates, and for being able to produce a wide variety of bioactive compounds derived from their secondary metabolism [1,9]. For example, genus *Aurantiochytrium* produces large amounts of PUFAs (>30% of total fatty acids), particularly docosahexaenoic acid (DHA, 22:6 *n*-3) and docosapentaenoic acid (DPA, 22:5 *n*-3). Therefore, the use of these microorganisms has been widely proposed as a strategy to develop functional foods and feeding technology. Although many studies have shown that these protists are efficient in producing and accumulating PUFAs [16,17,18,19], the quantity produced is still not enough to cover the needs of the aquaculture industry [7,19]. Therefore, it is still necessary to optimize the production process in order to enrich the content of PUFAs in protists.

Many eukaryotic-associated bacteria have been described in different sciences as effective enhancers of the growth and performance of their associated eukaryotic organisms. Among them, lactic acid bacteria (LAB) are considered a versatile and widespread group of beneficial microorganisms able of both enhancing plant growth and providing benefits for animal and human health. LAB are Generally Recognized As Safe (GRAS) by the U.S. Food and Drug Administration (FDA) and, therefore, are often used as ‘food additives’ of functional foods for their prebiotic and probiotic properties [20]. LAB are also recognized prebiotics and probiotics, capable of stimulating humoral and cellular immune responses. The genus *Lactobacillus*, *Pediococcus*, *Lactococcus*, *Carnobacterium* and *Leuconostoc* have been used as prebiotics in aquaculture systems due to their beneficial effect within the gastrointestinal tract [21]. 

In addition, several LAB strains have been characterized as plant growth-promoting bacteria (PGPB) and used as plant bioinoculants. This is due to most LABs producing auxins, such as the indole–3–acetic acid (IAA), which is a phytohormone that induces elongation of the plant cell and significantly improves the growth, yield, and stress response of some important crops. This effect on plants has been observed both when applying the phytohormone directly or through the inoculation of phytohormone-producing bacteria. Interestingly, this phytohormone can also significantly influence the production of biomass and the accumulation of fatty acids in different marine microorganisms. Indeed, it has been observed that IAA at concentrations of 10^−5^ M in *Scenedesmus obliquus* cultures increases microalgal growth up to 1.9 times and raises the content of fatty acid methyl esters [22]. 

Different bacteria have been reported with the ability to interact with microalgae and generate changes at the physiological and cellular levels. Through the interaction between volatile compounds, such as 2,3–butanediol and acetoin produced by *A. brasilense* strain 520, growth is promoted in the microalga *Chlorella vulgaris*. This microalga is capable of increasing its growth rate 6 fold and its volume 3 fold, in addition to showing an increase in the concentration of total lipids, total carbohydrates, and chlorophyll a [23].

In this way, the application of IAA could be an interesting strategy to increase the biomass and production of fatty acids by marine protists. However, the direct application of IAA can have a high cost and potential environmental impact that is yet to be determined. Consequently, the use of LABs as a safe source of IAA is proposed as a sustainable strategy for optimizing the production of biomass and PUFAs accumulation in the protist *Aurantiochytrium* sp. In this way, the main objective of this study was to evaluate if through the co-cultivation of the IAA-producing LAB and the protist *Aurantiochytrium* sp. strain T66 (ATCC PRA–276) it could be possible to induce the bio-accumulation of essential PUFAs in the protist.

## 2. Results

### 2.1. Isolation of L-Tryptophan-Dependent Auxin Lactic Acid Bacteria 

Samples isolated from three different sources (kefir, tomato rhizosphere (*Solanum lycopersicum*) and three intestines of juvenile rainbow trout) were analyzed in order to obtain LAB. Of the total bacteria isolated, 28 strains were considered putative LAB, mainly due to growth in the specific MRS and Gram-positive medium. 

The same 28 isolates obtained from different sources were analyzed for L-tryptophan-dependent auxin production by the Salkowski spectrophotometric method [24]. Six isolates were positive for IAA production (K6.10, R4.25, K4.5, K4.5, L, E1). All the strains were isolated from kefir except the R4.25 strain which was isolated from tomato rhizosphere. The IAA levels ranged from 4.2–5.8 (µg mL^−1^), with *Azospirillum brasilense* B6 strain being used as a reference for IAA production (Table 1) had higher levels than that found in the isolated bacteria (9.9 µg mL^−1^). The *Schleiferilactobacillus harbinensis* strain K4.5 and *L. kefiri* strain K6.10 were also positive for IAA production (4.8 and 5.8 µg mL^−1^, respectively). Furthermore, as these six strains were all capable of growing in 2673M2 medium, they were used for co-culture assays.

### 2.2. Protist, Culture Condition and Antagonism Test

The culture medium used in this study (2673) is essentially for the cultivation of the protist; hence at the time of generating the selection of the lactic acid bacteria to be evaluated, it was necessary that they could all grow in this medium. The six auxin-producing lactic acid bacteria evaluated can indeed grow in this 2673 medium. It is also important to highlight that two of these microorganisms, K4.5 and K6.10 strains, did not reveal an incompatibility with *Aurantiochytrium* sp. T66, as indicated by the absence of an inhibition halo in the compatibility test performed.

### 2.3. Identification of L-Tryptophan-Dependent Auxin Lactic Acid Bacteria 

The two best candidates for LAB were positively identified, namely strain K4.5 and K6.10. The sequences were compared in the NCBI database to identify similarity (Figure 1), with the strains being identified as LAB, *Lentilactobacillus kefiri* and *Schleiferilactobacillus harbinensis* (K6.10 with 100% identity and K4.5 with 99% identity, respectively). 

### 2.4. Total Biomass in Co-Culture

In Table 2, a two-way ANOVA analysis shows that the interaction between factors “bacteria” and “inoculation time” is significant, which shows that for all the parameters evaluated (total biomass, lipids, fatty acid profile) this interaction had a significant effect. Variations in total biomass (g/L) production were observed when bacteria interact with the protist Aurantiochytrium sp. T66 (Table 3). The final biomass obtained by co-culture after 96 h was significantly higher when the bacterium L. kefiri strain K6.10 was inoculated 24 h after the start of the culture (*p* < 0.05). After 144 h, it was observed that the interaction between L. kefiri strain K6.10 and *Aurantiochytrium* sp. T66 did not affect total biomass, upon inoculating the bacterial strain at 24 and 72 h from the start of the culture (2.11 and 2.23 g·L^−1^).

### 2.5. Analysis of Fatty Acid Profile in Co-Culture

Total lipid content per gram of biomass indicated that when the protist interacts with the bacteria through co-culture, a decrease in the total lipid content was generated. A decrease in lipid content was noted for all co-culture treatments studied (Figure 2A). Among these, the lowest observed yields were 35.0 and 30.4 mg g^−1^ of biomass (at 96–144 h, respectively), which correspond to the interaction with the bacteria *A. brasilensis* strain B6 inoculated at the start of the cultivation of *Aurantiochytrium* sp. T66 (T1). On the other hand, the highest yields were 102.1 and 84.5 mg g^–1^ of biomass (at 96 and 144 h, respectively), which correspond to the interaction between the protist and the bacterium *L. kefiri* strain K6.10 inoculated 72 h after the start of the culture. 

Even though total lipid contents decreased (Table 4), the yields of PUFAs increased dramatically when *Aurantiochytrium* sp. strain T66 was co-cultured with bacteria (Figure 2B). This rise was typically observed for all treatments 96 h from the start of the culture. *Lentilactobacillus kefiri* strain K6.10 inoculated 72 h after the start of the culture was the only treatment in which the total content of PUFA increased significantly at both 96 h and 144 h (22.3 and 30.9 mg g^−1^ of biomass, respectively) in comparison with the control (*p* < 0.05). The opposite case was registered with the co-inoculation of *A. brasilensis* B6 and *Aurantiochytrium* sp. T66 (T1); in this treatment, the lowest PUFA yields were observed, corresponding to just 3.6 and 3.1 mg g^−1^ of biomass (at 96 h and 144 h, respectively). On the other hand, regarding the K4.5 strain, at 96 h it yielded a PUFA content of 28.71 mg g^−1^, then it dropped to 13.88 mg g^–1^ at 144 h.

The DHA content increased significantly (*p* < 0.05) (20.9 mg g-q biomass) compared to the control (5.7 mg g^−1^ biomass) when the K.5 strain was inoculated at 0 h after the start of the culture (values recorded at 96 h of culture). On the other hand, at 144 h of culture, the DHA content (19.5 mg g^−1^ biomass) increased significantly when strain K6.10 was inoculated 72 h after the start of culture (Figure 2C) (*p* < 0.05).

ARA increased significantly (0.5 mg g^−1^ biomass) compared to the control (0.12 mg g^−1^ biomass) (*p* < 0.05) when strain K6.10 was inoculated at 0 h after starting the culture and measured at 96 h. It should be highlighted that the strain k6.10 inoculated at 72 h produces 0.62 mg g^−1^ biomass, which is significantly higher than the control (0.12 mg g^−1^ biomass) (Figure 2D), measurement of the 144 h of culture.

In relation to EPA concentration when the K4.5 strain was inoculated at 0 h after the start of the culture, a value of 1.16 mg g^−1^ biomass was obtained, being significantly different from the control (0.28 mg g^−1^) at 96 h of culture (*p* < 0.05). On the other hand, the K6.10 strain measured at that same time (96 h) promoted a significant increase (1.24 mg g^−1^ biomass) when inoculated at 0 h in comparison to the control (0.28 mg g^−1^ biomass). In addition, the k6.10 strain inoculated at 24 h induced an EPA concentration of 1.21 mg g^−1^.

The point of highest concentration of EPA (1.37 mg g^−1^ biomass) was related to the inoculum of the K6.10 strain at 72 h after the start of the culture, measured at 144 h, while the control value under these conditions was 0.37 mg g^−1^ (Figure 2E).

The highest DPA content recorded (5.9 mg g^−1^ biomass) was obtained with the inoculum of strain K4.5 72 h after the start of the culture, compared to the control that yielded 2.4 mg g^−1^ at 96 h. In addition, the measurement at 144 h highlights significant differences with the control (2.1 mg g^−1^ biomass) when strain K6.10 was inoculated 72 h after the start of the culture (8.1 mg g^−1^ biomass) (Figure 2F) (*p* < 0.05).

The Principal Component Analysis (PCA) for 96 h after starting the culture (Figure 3A) explained 78.3% of the total variance (Appendix A), where the first component (44.8%) is associated with the PUFA profile (PUFA, DHA, ARA, EPA, 20:5 *n*-3), DPA content and pH, and the second component (33.5%) was associated with total lipids, saturated fatty acid (SAFA) and monounsaturated fatty acid (MUFA) content. Direct significant correlations with the profile of PUFAs were observed in several cases: PUFA (r = 0.6), DHA (r = 0.6), ARA (r = 0.5), EPA (r = 0.4) and DPA content (r = 0.6) (Table 5). An inverse correlation was noted between IAA production and pH (r = −0.5), biomass (r = −0.4), lipids (r = −0.4) and SAFA (r = −0.4), as well as an inverse correlation between pH and MUFA (r = −0.8), PUFA (r = −0.4), DHA (r = −0.4), ARA (r = −0.4), and EPA (r = −0.4) content, 96 h after initiation of the culture. These results highlight that the treatments evaluated (bacteria and inoculation time) after 96 h of culture promote a positive relationship with the increase in total PUFA, particularly DPA, DHA, ARA and EPA.

The same PCA analysis performed on the data obtained 144 h after starting the culture (Figure 3B), explained 72.8% of the total variance (Appendix A). The first component (44.6%) was associated with the PUFA profile and pH, and the second component (28.2%) was associated with total lipids, SAFA and MUFA content. IAA production showed a significant direct correlation with pH (r = 0.4), while pH had a significant inverse correlation with total PUFA content (r = −0.5) and individual PUFAs [DHA (r = −0.6), ARA (r = −0.4), EPA (r = −0.6), DPA (r = −0.4) (Table 5)]. The inverse interaction between ARA content and pH was in agreement with the score plot (Figure 3B), where the co-culture treatment with *L. kefiri* strain K6.10 is distributed between quadrants II and III, quadrants that were associated with total PUFA content and the evaluated PUFA profile. The axenic culture of *Aurantiochytrium* sp. T66 was distributed in quadrant I, associated with SAFA content and biomass production. 

## 3. Discussion

The growth conditions of marine microorganism cultures can regulate the accumulation of PUFAs. The concentration of salts, temperature, light intensity and growth phase are also factors that are frequently used in manipulating the accumulation of PUFAs [24]. Therefore, the modification of culture growth conditions turns out to be a common strategy to stimulate the bioaccumulation of lipids [25,26,27]. For this reason, the present exploratory study evaluated protist–LAB co-cultures as a strategy to induce bioaccumulation of PUFAs in the protist *Aurantiochytrium* sp. T66.

In this context, the IAA phytohormone can significantly influence biomass production and accumulation of fatty acids in different microalgae [22]. For example, it has been observed that IAA at concentrations of 10^−5^ M in *S. obliquus* cultures can increase microalgal growth up to 1.9 times and boost the content of fatty acid methyl esters [22]. In our study, the isolated lactic acid bacteria operated in co-culture with no significant changes in biomass production. However, a higher PUFA content was obtained without changes in terms of the final biomass being obtained (Figure 2). In addition, it should be noted that the co-culture treatment with the K6.10 strain inoculated at 72 achieved the best results in terms of total PUFA and ARA contents. In particular, this treatment achieved 1.8 times more auxins than the control (Au) (Table 3); however, it should be noted that no correlation was found between the production of auxin and the increase in PUFA (Table 5), in contrast with literature [25,26,27]. All these results indicate that the specific co-culture treatment that gives the best results in PUFA accumulation is that of the K6.10 strain inoculated at 72 h.

On the other hand, statistically significant variations (*p* < 0.05) in pH were reported (Table 3). Considering all these factors, such treatment stands out as the one that gave the highest content of PUFA obtained (Figure 2).

Regarding co-cultures, it has been reported that the *Pelagibaca bermudensis* strain KCTC 13073BP can stimulate 2 fold the cell density of the microalga *Tetraselmis striata*, and that the total productivity of biomass and lipids can be substantially promoted by bacterial inoculation in the medium [28]. *Rhizobium* sp. can exert a mutualistic effect by interacting by co-culture, where the microalgae *Chlamydomonas reinhardtii*, *Chlorella vulgaris* and *Scenedesmus* sp. show increases in their growth under co-culture conditions, reaching an increase in growth ratios of up to 110%, in the case of *Chlorella vulgaris* [29]. These results are in contrast with our study on which the co-cultures show that the time of inoculation is an important factor when obtaining bioactive compounds of interest. Figure 2 shows how the final biomass obtained varies with the timing of bacteria inoculation. For example, in the culture of *Aurantiochytrium* sp. T66, in the case of co-culture with the bacteria *L. kefiri* K6.10, there is a significant decrease in biomass at the end of the culture (1.09 g L^−1^) compared to the control without bacteria (2.62 g L^−1^) when the bacteria are inoculated at the beginning of the culture (0 h) and to the extent that the bacteria is integrated at times after the beginning of the culture (24 and 72 h) the differences found are not significant. Other treatments generated a significant decrease in the biomass at the end of the culture, in all the inoculation times tested. These changes could be attributable to multiple factors, such as availability of nutrients, changes in pH, dissolved oxygen, among others.

The content and profile of fatty acids in Thraustochytrids can be influenced by the pH in the culture medium, and the decrease in PH is attributable to LAB effect [30]. For instance, in *Aurantiochytrium limacinum* strain PKU#SW8, pH 4.0 and pH 6.47 were optimal for achieving the maximal content and yield of DHA, respectively [18]. In the co-culture treatments, it was observed that pH decreased when the bacteria were inoculated, decreasing significantly from an initial pH of 7.02 to 6.3 in the co-cultures with the K6.10 strain (96 h after the start of the culture). The results observed in this study respond to various dynamics that can occur in a co-culture system, namely chemical factors such as changes in the C/N ratio, pH variations and presence and concentration of phytohormones, as these can generate changes in the profile of fatty acids and the accumulation of intracellular lipids.

In protists of the genus *Aurantiochytrium*, there are two pathways of fatty acid synthesis: the type I fatty acid synthase (FAS) pathway which starts producing saturated fatty acids, and then, through desaturation and elongation processes of the carbon chain, synthesizes PUFA of the *n*3 and *n*6 series, and the polyketide synthase-like (PKS) pathway which is involved in the synthesis of DHA (C22:6) and DPA (C22:5) [31,32]. Exploring the molecular mechanisms responsible for the establishment of symbiotic associations has a great advantage for the improvement of co-cultures at an industrial scale [33]. The accumulation of fatty acids associated with the FAS pathway indicates that there is a stimulation of this pathway, where pH in combination with auxins can play an important role in the regulation of both biosynthesis pathways under the conditions evaluated. This effect should corroborate different concentrations of the precursor L-tryptophan and correlated with gene expression in both pathways.

## 4. Materials and Methods

### 4.1. Isolation of L-Tryptophan-Dependent Auxin Lactic Acid Bacteria 

Lactic acid bacteria were obtained by isolation from 3 different sources, a sample of kefir [34,35], a sample of tomato rhizosphere (*Solanum lycopersicum*) and 3 intestines of juvenile rainbow trout [36]. The kefir granules were previously pre-incubated in 25 mL of milk for 48 h prior to isolation, maintained at a temperature of 25 °C and, after pre–incubation, a serial dilution was performed in 0.85% saline solution in a 1:10 ratio. For the tomato rhizosphere sample, 1 g of soil was weighed from the roots of a plant maintained under greenhouse conditions and diluted in a 1:10 ratio of 0.85% saline solution. Juvenile rainbow trout were sacrificed with an overdose of benzocaine, whole intestines were taken aseptically, intestines were weighed and diluted in 0.85% saline solution in a 1:10 ratio. For the isolation of lactic acid bacteria, serial dilutions were employed, 10–4, 10–5 and 10–6 dilutions were considered for all the samples analyzed. Culture medium plates were inoculated with MRS agar medium (Man, Rogosa and Sharpe), with the plates being incubated at 25 °C. After 96 h, isolation and purification of grown colonies was started. After the isolation and purification of bacterial strains, a morphological analysis of the strains obtained was performed by Gram staining and subsequent visualization by microscopy; this analysis was used as the first selection criterion for lactic acid bacteria, considering bacterial morphology and Gram positive.

For auxins production, strains were grown in MRS broth for 96 h, and 100 μL aliquots were transferred to 10 mL of YPG medium (1% yeast extract, 2% peptone and 2% glucose) supplemented with 300 μg mL^−1^ of L–tryptophan. The tubes were then incubated at 25 °C and 120 rpm on an orbital shaker and samples were taken 120 h after inoculation. Optical density (OD) at 560 nm was recorded as an indicator of growth and an aliquot from each flask was centrifuged (10,000 rpm) to remove bacterial cells. One milliliter of supernatant was mixed with 1 mL of Salkowski’s reagent (150 mL of 18 M H_2_SO_4_, 250 mL of AD, 7.5 mL of 0.5 M FeCl_3_–6H_2_0) and the absorbance at 535 nm was measured after 25 minutes’ incubation [24]. Auxin concentration was estimated from a standard curve with IAA (Sigma I–2886) and expressed in micrograms per milliliter. Auxin production was measured in the same way in the co-cultured assays.

### 4.2. Protist, Culture Condition and Antagonism Test

The protist *Aurantiochytrium* sp. strain T66 (ATCC PRA-276), was obtained from the ATCC collection (American Type Culture Collection, USA). The strain was activated and maintained in 2673 modified medium containing 2% glucose, 1% peptone and 0.1% yeast extract in sea water. The components of the culture medium were diluted in “sea water” containing: 27.12 g L^−1^ NaCl, 5.23 g L^−1^ MgCl_2_–6H_2_O, 6.77 g L^−1^ MgSO_4_–7H_2_O, 0.15 g L^−1^ CaCl_2_–2H_2_O, 0.73 g L^−1^ KCl, 0.20 g L^−1^ NaHCO_3_, with the medium being autoclaved for 21 min at 121 °C. 

Of the positive bacteria for auxin production, the detection of antagonism with the protist *Aurantiochytrium* sp. T66 was performed using the diffusion method on Whatman microfiber filter discs. In bales with 2673 medium, 100 μL of protist culture was seeded by sweeping in 2673 broth with 96 h of growth, with 3 whatman filter discs (MF Millipore, Darmstadt, Germany) being placed per plate (9 mm in diameter), where each bacterium was inoculated; 20 μL of each bacteria evaluated were inoculated in the corresponding disks and subsequently incubated at 25 °C. Antagonism was evaluated by observing the inhibitory zones around the disk produced at 144 h from incubation. Each assay was performed in triplicate. The degree of antagonism shown was determined by measuring the mean diameter of the clear zone of inhibition: −, no inhibition (<1 mm); +, weak inhibition (<5 mm); ++, mild inhibition (=5 mm); and +++, strong inhibition (>10 mm).

### 4.3. Identification of L–Tryptophan-Dependent Auxin Lactic Acid Bacteria 

The strains of lactic acid bacteria were identified at the molecular level. The partial 16s rRNA gene was amplified by PCR using universal primers [37]. The sequences were analyzed with the MEGA7 software and compared with the NCBI database. The strain was registered in the NCBI database. Sequences were entered into the NCBI database under the access codes ON734020 (*S. harbinensis* strain K4.5) and ON734021 (*L. kefiri* strain K6.10).

### 4.4. Co-Culture Design 

The growth kinetics of the 180 h protist in the 2673 medium indicates that total lipids increase in the exponential phase from 20 to 72 h of culture, with total lipid content decreasing, but PUFA accumulation beginning. For this reason, in the co-culture assay, the fatty acid profile was determined only after 96 and 144 h of beginning the trials (Appendix A). Thus, to determine the effect of bacteria on the biomass and the accumulation of fatty acids in the protist, a co-culture was carried out in the 2673 medium (0.1% yeast extract, 1% peptone and 2% glucose, *w/v* of sea water). The protist *Aurantiochytrium* sp. T66 was cultiveted together with the bacteria *L. kefiri* strain K6.10 and *S. harbinensis* strain K4.5, on the other hand, the bacteria *Azospirillum* sp. strain B6 was used as a reference for auxin production, all the strains were activated for 24 h at 25 °C. After activation, the strains were conditioned in 100 mL of 2673 media for 96 h at 25 °C, from conditioning 8 mL of each strain were taken and flasks were inoculated with 400 mL of 2673 media, where they were kept under constant agitation at 120 rpm at a temperature of 25 °C. This culture was used as inoculum for the co-culture assay. Before preparing the inoculum, the concentration of the different strains employed was determined using a Neubauer chamber, resulting in T66 having a concentration of 1 × 10^6^, B6 of 1 × 10^7^, K6.10 of 9.7 × 10^6^, and K4.5 of 8 × 10^6^.

*Aurantiochytrium* sp. T66 was evaluated under different conditions in 2673 medium supplemented with L–Tryptophan (300 μg mL^−1^), 3 inoculation times were performed for each of the bacteria, where:

Au: *Aurantiochytrium* sp. strain T66 (control); B: co–culture of *Aurantiochytrium* sp. T66 together with *A. brasilensis* strain B6; C: co–culture of *Aurantiochytrium* sp. strain T66 together with *S. harbinensis* strain K4.5; D: co–culture of *Aurantiochytrium* sp. strain T66 together with *Lentilactobacillus kefiri* strain K6.10; T: inoculation time of the bacteria in the culture of the protist *Aurantiochytrium* sp. strain T66 (T1: 0 h, T2: 2 4 h and T3: 72 h).

### 4.5. Analysis of Fatty Acid Profiles in Co–Culture 

Total biomass was determined at 96 and 144 h, with a 10 mL aliquot of the culture medium from each flask being filtered using a previously weighed glass microfiber filter paper (GF/C:1.2 m, Whatman, Germany). The biomass retained on the filter was washed twice with distilled water and dried in an oven at 105 °C for 24 h. The biomass content was calculated as the difference between the initial weight and the final weight.

Total lipids were extracted according to the modified Folch method [38], taking 0.1 g of sample and using a chloroform:methanol (2:1 and 0.01% BHT) solution as extractant, in addition to 1.5 mL of 0.1 N HCL and 1 mL of 0.5% MgCl. The solution was then vortexed, centrifuged at 3500 rpm and the chloroform phase worked on, bringing it to dryness and constant weight under a stream of nitrogen. The value obtained was expressed as a percentage of total lipids.

To obtain fatty acid methyl esters FAME, the total lipids are methylated according to the methods of Morrison and Smith [39]. For this, 14% Boron Trifluoride in methanol was added to the lipids, with the tubes then being boiled in a water bath for 20 min; subsequently, the tubes were left at room temperature and 3 mL of hexane and 1.5 mL of ultra-pure water was added, with this mix then being vortexed and centrifuged. The hexanic phase was brought to dryness under a current of nitrogen and the obtained mg of lipids were diluted in dichloromethane and transferred to vials. In this way, the methyl esters were ready to be injected, with 1 µL being used; an Agilent 768b autosampler attached to a Hewlett Packard model HP6890 GC chromatograph equipped with a flame ionization detector (FID) was employed, with He being used as the carrier gas. The separation was performed by using a Supelco SP2380 Capillary Column (30 m length × 0.25 internal diameter × 0.20 µm film thickness). The injector and detector temperatures were set at 220 and 200 °C, respectively. To ensure the best possible separation of fatty acids, the following schedule was used: 60 °C for 1 min, followed by 4 °C per minute at 204 °C, and finally 2 °C per minute at 220 °C and stabilization for 2 min. Fatty acids were identified by comparison with standard Supelco-37 fatty acids (Sigma-Aldrich, Darmstadt, Germany CRM47885) and quantified in HPCHEM Stations software (Agilent Technologies, Santa Clara, CA, USA), being expressed as their percentage area based on total identified fatty acids.

### 4.6. Statistical Analysis

Prior to statistical analyses, all data were checked for normality using a Shapiro–Wilk test and homogeneity of variance using a Bartlett’s test. To investigate the existence of significant differences (*p* < 0.05) between the different conditions studied, a two-way analysis of variance (two-way ANOVA) using co-culture treatment and bacteria inoculation time as fixed factors was carried out, followed by Tukey’s post hoc test, the Shapiro–Wilk test, Bartlett’s test, two-way ANOVA test and Tukey’s post hoc test was carried out with R software version 3.6.3 [40]. Principal component analysis (PCA) was used to visualize correlation and the relationship between treatments and the variables being evaluated, with the software PRIMER v6 (PRIMER–E Ltd.) being used to perform this analysis. Pearson’s correlation was performed using the SPSS software version 26 (IBM, Armonk, NY, USA).

## 5. Conclusions

This study demonstrates that the use of LAB as an inoculant in co-cultures with protist is an alternative and promising strategy to effectively improve protist’s production of total PUFA. The *Lentilactobacillus kefiri* K6.10 strain inoculated at 72 h gives the best PUFA content (30.89 mg g^−1^ biomass) measured at 144 h of culture, three times higher than the control (8.87 mg g^−1^ biomass). Therefore, this effect can be used to further optimize the co-culture strategy in order to continue to modify the profiles of PUFA. 

The interaction between the protist and bacteria leads to changes in fatty acid profiles. This effect is associated with the inoculation time of each bacterium and the type of strain in each co-culture treatment. Co-culture can lead to the generation of complex biomasses with higher added value for developing aquafeed supplements.

## Figures and Tables

**Figure 1 marinedrugs-21-00142-f001:**
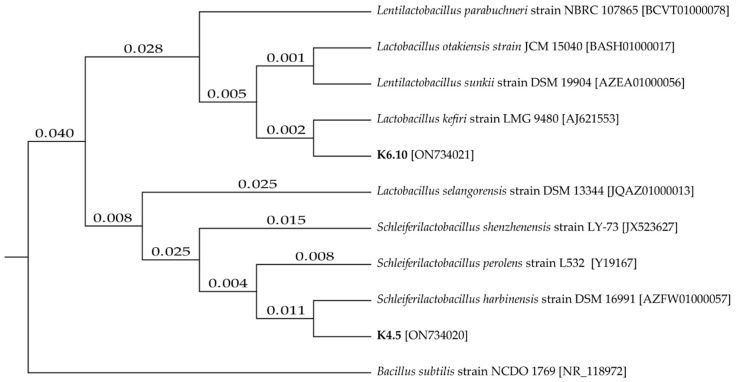
Tree of phylogenetic distances for isolates K4.5 and K6.10 based on the neighbor–joining method.

**Figure 2 marinedrugs-21-00142-f002:**
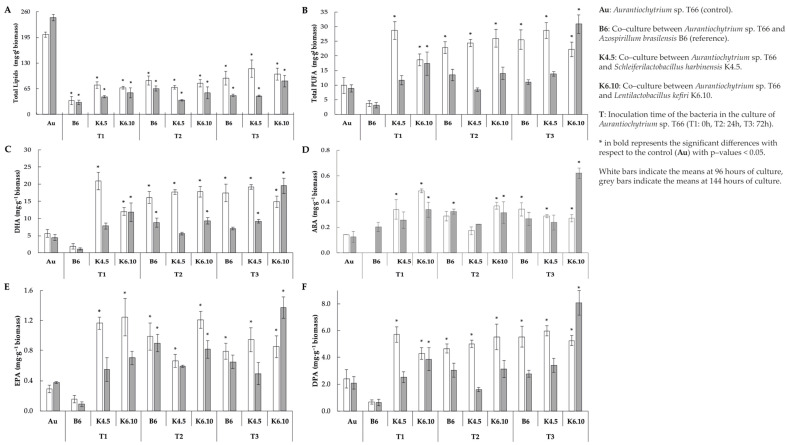
Profiles of biomass and fatty acids profile of extracted lipid from co–culture assay (**A**) Total lipids (**B**) Total polyunsaturated fatty acid, (**C**) docosahexaenoic acid, (**D**) arachidonic acid, (**E**) EPA = eicosapentaenoic acid and (**F**) docosapentaenoic acid yield at 96 and 144 h of culture.

**Figure 3 marinedrugs-21-00142-f003:**
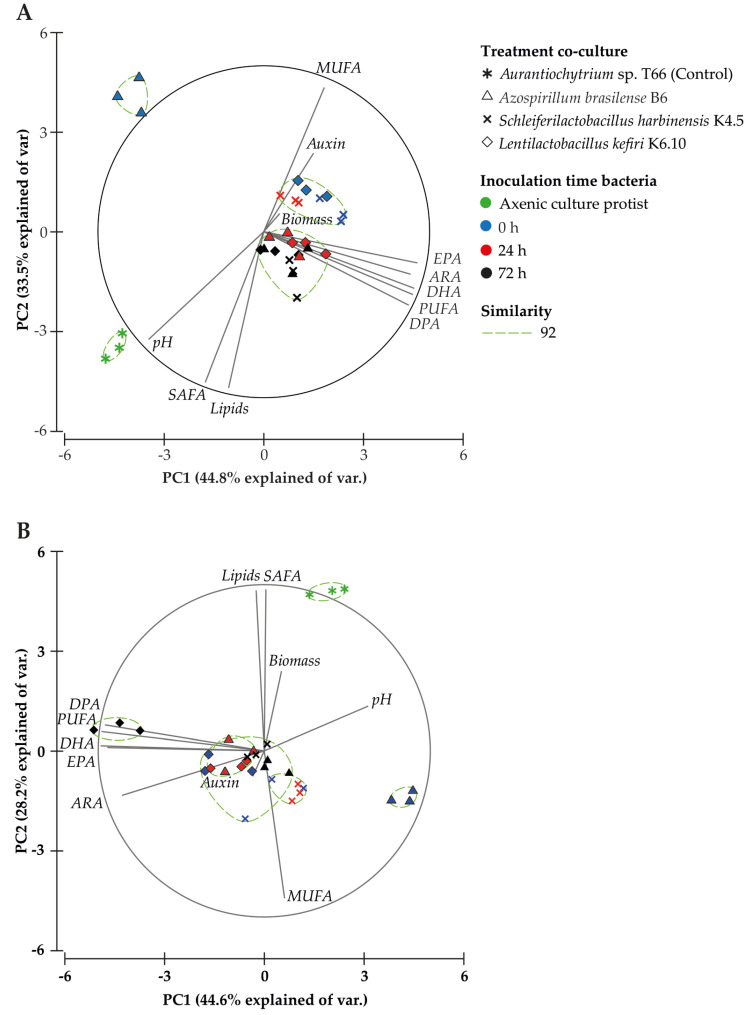
Principal Component Analysis (PCA) based on treatment, fatty acid profile and biochemical analyses at 96 (**A**) and 144 h (**B**) after starting the culture. The similarity test grouped treatments by Bray–Curtis similarity.

**Table 1 marinedrugs-21-00142-t001:** Summary of production of indole–3–acetic acid (IAA) of the strains evaluated in this study.

		Auxin (µg mL^−1^)
Control		
	*Aurantiochitrium* sp. *T66*	10.0 ± 3.4
Reference		
	*Azospirillum brasilensis B6*	9.8 ± 2.8
Isolated		
	*Schleiferilactobacillus harbinensis K4.5*	4.8 ± 0.2
	*Lentilactobacillus kefiri K6.10*	5.8 ± 0.4

**Table 2 marinedrugs-21-00142-t002:** Resume of two-way ANOVA performed on the effect of co-culture with bacteria (CoB) and inoculation time (IT) on total lipids and the fatty acid profile at 96 and 144 h (h) of culture.

		96 h of Culture					144 h of Culture				
Parameter	Item	*DF*	*SS*	*MS*	*F*	*p*	*DF*	*SS*	*MS*	*F*	*p*
Total lipids	CoB	3	29,084	9695	63.503	**<0.001** *	3	103,917	34,639	453.90	**<0.001** *
	IT	2	8781	4391	28.76	**<0.001** *	2	1255	627	8.22	**<0.001** *
	CoB X IT	4	2839	710	4.65	**0.009** *	4	2616	654	8.57	**<0.001** *
	Residuals	19	2901	153			20	1526	76		
Biomass	CoB	3	0.80	0.27	1.21	0.33	3	2.29	0.76	20.64	**<0.001** *
	IT	2	0.39	0.20	0.90	0.43	2	1.45	0.73	19.64	**<0.001** *
	CoB X IT	4	0.59	0.15	0.67	0.62	4	2.66	0.67	17.97	**<0.001** *
	Residuals	18	3.94	0.22			19	0.70	0.04		
Total SAFA	CoB	3	39,729	13,243	166.79	**<0.001** *	3	116,745	38,915	937.84	**<0.001** *
	IT	2	7348	3674	46.27	**<0.001** *	2	1003	502	12.09	**<0.001** *
	CoB X IT	4	1414	354	4.45	**0.010** *	4	1508	377	9.09	**<0.001** *
	Residuals	19	1509	79			20	830	41		
Total MUFA	CoB	3	216.95	72.32	6.53	**0.004** *	3	197.86	65.95	9.83	**<0.001** *
	IT	2	217.18	108.59	9.80	**0.001** *	2	18.12	9.06	1.35	0.284
	CoB X IT	4	25.08	6.27	0.57	0.691	4	108.08	27.02	4.03	**0.017** *
	Residuals	17	188.32	11.08			18	120.83	6.71		
Total PUFA	CoB	3	727.50	242.49	40.02	**<0.001** *	3	748	249.35	61.45	**<0.001** *
	IT	2	385.50	192.76	31.81	**<0.001** *	2	324.60	162.28	39.99	**<0.001** *
	CoB X IT	4	586.90	146.73	24.21	**<0.001** *	4	373.90	93.48	23.04	**<0.001** *
	Residuals	19	115.10	6.06			20	81.20	4.06		
ARA	CoB	3	0.21	0.07	37.36	**<0.001** *	3	0.27	0.0913	35.24	**<0.001** *
	IT	2	0.00	0.00	0.93	0.411	2	0.06	0.0299	11.54	**<0.001** *
	CoB X IT	4	0.31	0.08	41.58	**<0.001** *	4	0.14	0.0345	13.31	**<0.001** *
	Residuals	19	0.04	0.00			20	0.05	0.0026		
DHA	CoB	3	428.70	142.89	54.73	**<0.001** *	3	361.30	120.44	68.90	**<0.001** *
	IT	2	188.80	94.40	36.16	**<0.001** *	2	127.90	63.93	36.57	**<0.001** *
	CoB X IT	4	326.40	81.59	31.25	**<0.001** *	4	159.10	39.79	22.76	**<0.001** *
	Residuals	19	49.60	2.61			20	35.00	1.75		
EPA	CoB	3	1.69	0.56	27.71	**<0.001** *	3	1.30	0.43	39.35	**<0.001** *
	IT	2	0.07	0.03	1.64	0.220	2	0.78	0.39	35.46	**<0.001** *
	CoB X IT	4	1.89	0.47	23.19	**<0.001** *	4	1.04	0.26	23.45	**<0.001** *
	Residuals	19	0.39	0.02			20	0.22	0.01		
DPA	CoB	3	28.10	9.367	31.71	**<0.001** *	3	47.18	15.727	52.13	**<0.001** *
	IT	2	19.87	9.936	33.64	**<0.001** *	2	31.66	15.83	52.47	**<0.001** *
	CoB X IT	4	24.24	6.06	20.52	**<0.001** *	4	26.54	6.635	21.99	**<0.001** *
	Residuals	19	5.61	0.295			20	6.03	0.302		

**CoB**: Co-culture treatment with bacteria; (**IT**) inoculation time of the bacteria. (*) In bold indicates differences significates (*p* < 0.05); DF: degree of freedom; SS: sum of square; MS: mean of square; F: likelihood ratio; P: probability. SAFA = saturated fatty acid; MUFA = monounsaturated fatty acid, PUFA = polyunsaturated fatty acid, ARA = arachidonic acid, DHA = docosahexaenoic acid; EPA = eicosapentaenoic acid, DPA = docosapentaenoic acid.

**Table 3 marinedrugs-21-00142-t003:** Biomass production, L-tryptophan-dependent auxins and pH from co-culture assay.

T		Biomass (g L^−1^)		Auxin (µg mL^−1^)		pH	
	Item	96 h	144 h	96 h	144 h	96 h	144 h
Control	Au	1.19 ± 0.2	2.62 ± 0.2	9.7 ± 2.3	10.8 ± 1.2	6.9 ± 0.0	6.9 ± 0.0
0 h	B6	1.43 ± 0.1	**1.82** ± **0.2** *	15.9 ± 3.0	15.8 ± 2.2	**6.3** ± **0.1** *	7.1 ± 0.1
	K4.5	1.32 ± 0.1	**1.38** ± **0.4** *	**22.6** ± **3.1** *	**17.8** ± **4.4** *	**6.2** ± **0.0** *	6.8 ± 0.1
	K6.10	1.43 ± 0.2	**1.09** ± **0.2** *	**22.4** ± **3.3** *	16.8 ± 1.1	**6.2** ± **0.0** *	**6.7** ± **0.1** *
24 h	B6	1.42 ± 0.3	**1.09** ± **0.2** *	8.9 ± 2.0	7.0 ± 2.9	**6.3** ± **0.0** *	**6.6** ± **0.0** *
	K4.5	0.93 ± 0.0	**1.73** ± **0.1** *	**32.0** ± **2.1** *	16.2 ± 1.8	**6.3** ± **0.0** *	**6.7** ± **0.0** *
	K6.10	**2.24** ± **0.1** *	2.11 ± 0.1	11.2 ± 2.5	7.1 ± 2.1	**6.4** ± **0.0** *	**6.5** ± **0.1** *
72 h	B6	1.60 ± 0.2	1.96 ± 0.2	8.7 ± 4.1	9.4 ± 3.2	**6.3** ± **0.2** *	6.8 ± 0.0
	K4.5	0.90 ± 0.1	**1.92** ± **0.3** *	14.5 ± 3.6	10.0 ± 0.2	**6.4** ± **0.1** *	**6.7** ± **0.0** *
	K6.10	0.90 ± 0.2	2.23 ± 0.0	15.4 ± 1.7	**19.1** ± **2.1** *	**6.3** ± **0.1** *	**6.7** ± **0.1** *

**Au**: *Aurantiochytrium* sp. T66 (control); **B6**: Co-culture between *Aurantiochytrium* sp. T66 and *Azospirillum brasilensis* B6 (reference); **K4.5**: Co-culture between *Aurantiochytrium* sp. T66 and *Schleiferilactobacillus harbinensis* K4.5; **K6.10**: Co-culture between *Aurantiochytrium* sp. T66 and *Lentilactobacillus kefiri* K6.10; **T**: Inoculation time of the bacteria in the culture of *Aurantiochytrium* sp. T66. ***** in bold represents the significant differences with respect to the control (Au) with *p*–values < 0.05. Values indicate means ± SD.

**Table 4 marinedrugs-21-00142-t004:** Polyunsatured fatty acid (PUFA) Yield and PUFA Profile of the Co-Cultivation Assay.

T		PUFA (mg·L^−1^)		DHA (mg·L^−1^)		ARA (mg·L^−1^)		EPA (mg·L^−1^)		DPA (mg·L^−1^)	
	Item	96 h	144 h	96 h	144 h	96 h	144 h	96 h	144 h	96 h	144 h
Control	Au	10.7 ± 2.6	23.1 ± 1.1	6.1 ± 1.0	11.5 ± 1.2	0.1 ± 0.1	0.3 ± 0.1	0.3 ± 0.1	1.0 ± 0.1	2.6 ± 0.6	5.4 ± 0.7
0 h	B6	5.4 ± 1.9	**5.5** ± **0.8** *	2.7 ± 1.3	**2.0** ± **0.3** *	0.0 ± 0.0	0.4 ± 0.1	0.2 ± 0.1	**0.2** ± **0.0** *	1.0 ± 0.3	**1.1** ± **0.3** *
	K4.5	**37.8** ± **2.5** *	17.7 ± 1.1	**27.5** ± **2.2** *	12.1 ± 0.5	**0.4** ± **0.1** *	0.4 ± 0.1	**1.5** ± **0.0** *	0.8 ± 0.2	**7.5** ± **0.8** *	3.7 ± 0.1
	K6.10	26.6 ± 4.5	17.8 ± 5.7	17.1 ± 2.4	12.1 ± 4.0	**0.7** ± **0.1** *	0.3 ± 0.1	**1.8** ± **0.4** *	0.7 ± 0.2	6.1 ± 1.2	3.9 ± 1.3
24 h	B6	**35.1** ± **10.0** *	14.9 ± 4.2	**24.8** ± **7.5** *	9.8 ± 2.8	**0.4** ± **0.1** *	0.4 ± 0.1	**1.5** ± **0.5** *	1.0 ± 0.3	**7.1** ± **1.8** *	3.4 ± 1.0
	K4.5	23.3 ± 1.7	14.5 ± 1.4	16.9 ± 1.2	9.7 ± 0.9	0.2 ± 0.0	0.4 ± 0.0	0.7 ± 0.0	1.0 ± 0.1	4.8 ± 0.4	2.8 ± 0.3
	K6.10	**61.3** ± **8.7** *	29.6 ± 4.5	**41.4** ± **5.1** *	**19.7** ± **2.1** *	**0.9** ± **0.1** *	**0.7** ± **0.2** *	**2.7** ± **0.5** *	**1.7** ± **0.2** *	**13.6** ± **1.7** *	6.6 ± 1.3
72 h	B6	**40.3** ± **13.9** *	21.8 ± 3.5	**27.8** ± **10.3** *	13.9 ± 2.1	**0.5** ± **0.1** *	0.5 ± 0.1	1.3 ± 0.5	1.3 ± 0.3	**8.9** ± **3.3** *	5.4 ± 0.7
	K4.5	26.1 ± 6.0	26.7 ± 5.6	17.3 ± 2.8	17.8 ± 3.7	0.3 ± 0.0	0.4 ± 0.0	0.9 ± 0.3	0.9 ± 0.2	5.4 ± 1.0	6.6 ± 1.9
	K6.10	19.9 ± 2.6	**68.9** ± **6.8** *	13.2 ± 1.7	**43.6** ± **5.0** *	0.2 ± 0.0	**1.4** ± **0.1** *	0.7 ± 0.0	**3.1** ± **0.3** *	4.7 ± 0.6	**18.0** ± **1.9** *

**Au**: *Aurantiochytrium* sp. T66 (control); **B6**: Co–culture between *Aurantiochytrium* sp. T66 and *Azospirillum brasilensis* B6 (reference); **K4.5**: Co–culture between *Aurantiochytrium* sp. T66 and *Schleiferilactobacillus harbinensis* K4.5; **K6.10**: Co–culture between *Aurantiochytrium* sp. T66 and *Lentilactobacillus kefiri* K6.10; **T:** Inoculation time of the bacteria in the culture of *Aurantiochytrium* sp. T66. * in bold represents the significant differences with respect to the control (Au) with *p*-values < 0.05. Values indicate means ± SD.

**Table 5 marinedrugs-21-00142-t005:** Pearson’s correlation for all independent variables at 96 and 144 h (h) after starting the culture.

	96 h of Culture				144 h of Culture			
	*Treatment*	*Auxin*	*pH*	*Biomass*	*Treatment*	*Auxin*	*pH*	*Biomass*
**Variable**								
*Treatment*	1	−0.088	−0.265	−0.182	1	−0.116	−0.589 *	0.200
*Auxin*	−0.088	1	−0.509 *	−0.391 *	−0.116	1	0.403 *	−0.044
*pH*	−0.265	−0.509 *	1	−0.051	−0.589 *	0.403 *	1	0.204
*Biomass*	−0.182	−0.391 *	−0.051	1	0.200	−0.044	0.204	1
*Lipids*	0.018	−0.390 *	0.726 *	−0.214	−0.312	−0.119	0.262	0.339
*SAFA*	−0.032	−0.435 *	0.799 *	−0.220	−0.328	−0.183	0.259	0.338
*MUFA*	0.068	0.441 *	−0.821 *	0.119	0.150	0.097	−0.094	−0.289
*PUFA*	0.638 *	0.103	−0.372 *	−0.012	0.637 *	0.136	−0.512 *	−0.031
*DHA*	0.618 *	0.134	−0.399 *	−0.013	0.665 *	0.104	−0.612 *	−0.083
*ARA*	0.516 *	0.095	−0.443 *	0.164	0.616 *	0.258	−0.383 *	−0.154
*EPA*	0.467 *	0.114	−0.509 *	0.216	0.637 *	−0.048	−0.647 *	−0.100
*DPA*	0.667 *	0.071	−0.311	−0.033	0.654 *	0.128	−0.448 *	0.015

(*) indicate significance for pairwise correlations (*p* < 0.05).

## Data Availability

Not applicable.

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
