# Peer review of "Protist–Lactic Acid Bacteria Co-Culture as a Strategy to Bioaccumulate Polyunsaturated Fatty Acids in the Protist Aurantiochytrium sp. T66"

_marinedrugs, 2023, doi:10.3390/md21030142_

Round 1

Reviewer 1 Report

The paper by Marileo et al  “Protist–lactic acid bacteria co–culture as a strategy to bioaccu-mulate polyunsaturated fatty acids in the protist Aurantio-chytrium sp. T66.” describes  isolation of lactic acid bacteria, and studies of how cocultures of these bacteria with Auratiochytrium sp T66  influences the fatty acid profile of the bacterium. As the authors point out, thraustochytrids are good producers of marine fatty acids (omega-3), and may be an alternative to fish oil for aquaculture feed. They also point out that co-culture strategies have worked for microalgae to change their fatty acid profile. Hence, this is a path well worth exploring. However, the present manuscript/ study needs to be improved.

Major points:

1.       The abstract indicates that a higher PUFA production was achieved. However, from the conclusion it becomes clear that the increase was in the production of ARA, an omega6 PUFA. Given that the lack of PUFA for feed is on the omega-3 fatty acids, while plant oils provide sufficient amounts of omega 6 PUFAs for food and feed, the authors need to rewrite the abstract and discussion. That being said, if a study is well conducted and exploring a novel and logical path, it should not be left unpublished merely because the results were not those one hoped for.

2.       Table 2 is very large, could it be divided somehow? Perhaps some data could be presented as graphs, and the Table-data as supplementary?  What is the meaning of the asterixes?

3.       I am not able to evaluate the results as they are now, because I lack some data.

a.       Table 2: the amount of DHA is not shown in my copy. Given that that is the main PUFA produced by  the genus, the data must be added. (It is found in Fig 2, but for comparison)

b.       How much of the biomass is thraustochytrid, and how much is bacteria? (You probably plated to determine CFU and know the dry weight of the two to be able to provide an estimate). Or you could redo the growth experiment to get these parameters. Unless you know that you separated the two organisms before biomass was calculated. But I see no evidence for that in your Paper.

c.       How were the growth curves for thraustochytrid and for bacterium. When did the lipid accumulation phase start? The latter is important since it is known that the fatty acid profile for thraustochytrids are very dependent on their growth phase, including their preferentially using SFA for energy when starving and their phospholipids containing more PUFA then their TAG-fraction do. Hence, any difference in growth between the cultures might also have an impact on their fatty acid profile.

d.       Since this is about improving bioprocesses: providing the numbers for the different fatty acids as both g/L (and % of total lipid) is needed. Of course, when the total lipid is provided both as g/L and g/biomass, which you should do, it is possible to calculate the others, but as a reader I prefer the authors to analyze their data providing me with numbers that can be compared directly.

4.       The conclusion part should be rewritten

Minor points

5.       Abstract, L1: “preferably aquatic” – do you know of any non-aquatic thraustochytrid? “marine” might have been a correct word.

6.       Abstract L 2: last words are just not in the right order

7.       Abstract L4 Why is “of fish” added?

8.       Abstract L5: you did not test cocultures of protists in general, only one strain. Rewrite this and the following lines

9.       Abstract last line: “technology with high nutritional attributes”? I can guess what you mean, but you need to rewrite.

I have not gone as detailed through the sentences later in the manuscript, I suggest too much rewriting

10.   P2 Thraustochytrids bioaccumulate: Not the best word. Fish may bioaccumulate, thraustochytrids produce.

11.   P3 and throughout: the names of microorganisms are only written fully the first time, then the genus name is to be abbreviated to the initial.

12.   2.1: I would like to know the origin of the 6 you found – from which of your three sources? And why did you only sequence two of them. That might make sense if you picked the best two auxin producers. In that case, this information needs to be put after 2.2

13.   2.2 Line 4, should it be 5.8 and not 5.6?

14.   2.3 Why is that particular medium essential, there are many different media being used for this genus and even this strain? Indeed, you should probably explain why you chose this medium with a fairly low content of sugar and a relative high amount of nitrogen.

15.   I suggest that you reorganize the 2.1-2.3 to make the story smoother.

16.   2.4: another headline, this is not the methods section. And you can write it shorter: after 96 h, one treatment had a significantly higher biomass, and that was the only treatment that did not display a significant lower biomass after 144 h. Moreover, given that we are talking lipid accumulating organisms, would the increase between 96 and 144 h be lipids alone, or the total biomass?

17.   2.5: The Table shows two strain, only one is commented on in the first paragraph.

18.   The PCA analysis does not add anything to my understanding of the data. It would have made sense as a bridge to further growth experiments, e.g. if pH is that important, should you have tested growth of the thraustochytrid alone, but in a medium buffered to 6.2?

19.   Fig2: Grey and white is 96 and 144 h? and there are a couple of “y” in stead of “and”

20.   Discussion. L2 “to project its use” I do not understand the phrase in this context.  And feed supplements or a feed supplement

21.   P 9: Our results are due to the fact …, I do not understand this – and I do not think this paragraph comes with necessary information not already covered well in the introduction. And since your Table 3 shows that auxin is not the most important factor – is it relevant at all.

22.   The paragraph starting with “different bacteria” is a justification of the study. If it were to be in the Paper, it would fit better in the introduction

23.   P9 when discussing the biomass, please consider what would be the aim of your process. Biomass can be both lipids and the proteins (lipid free biomass). If lipids is the intended product, the amount of lipids is the interesting number.

24.   P10 top, is it possible that lactic acid bacteria could lower the pH?

25.   Material and methods seems to be sufficient, perhaps a bit more detailed than necessary.  P12 3 lines above 4.7: The identification of fatty acids will be : was?

26.   Supplementary Table 1 was not well shon in the manuscript – some of the right hand columns are missing.

Author Response

Response to Reviewer 1

The paper by Marileo et al  “Protist–lactic acid bacteria co–culture as a strategy to bioaccu-mulate polyunsaturated fatty acids in the protist Aurantio-chytrium sp. T66.” describes  isolation of lactic acid bacteria, and studies of how cocultures of these bacteria with Auratiochytrium sp T66  influences the fatty acid profile of the bacterium. As the authors point out, thraustochytrids are good producers of marine fatty acids (omega-3), and may be an alternative to fish oil for aquaculture feed. They also point out that co-culture strategies have worked for microalgae to change their fatty acid profile. Hence, this is a path well worth exploring. However, the present manuscript/ study needs to be improved.

A: Dear Reviewer, we are very grateful for all your comments, which helped us improve the quality of our work.

Major points:

  1. The abstract indicates that a higher PUFA production was achieved. However, from the conclusion it becomes clear that the increase was in the production of ARA, an omega6 PUFA. Given that the lack of PUFA for feed is on the omega-3 fatty acids, while plant oils provide sufficient amounts of omega 6 PUFAs for food and feed, the authors need to rewrite the abstract and discussion. That being said, if a study is well conducted and exploring a novel and logical path, it should not be left unpublished merely because the results were not those one hoped for.

 A: the abstract was rewritten

  1. Table 2 is very large, could it be divided somehow? Perhaps some data could be presented as graphs, and the Table-data as supplementary?  What is the meaning of the asterixes?

  1. I am not able to evaluate the results as they are now, because I lack some data.

  1. Table 2: the amount of DHA is not shown in my copy. Given that that is the main PUFA produced by  the genus, the data must be added. (It is found in Fig 2, but for comparison)
  2. How much of the biomass is thraustochytrid, and how much is bacteria? (You probably plated to determine CFU and know the dry weight of the two to be able to provide an estimate). Or you could redo the growth experiment to get these parameters. Unless you know that you separated the two organisms before biomass was calculated. But I see no evidence for that in your Paper.
  3. How were the growth curves for thraustochytrid and for bacterium. When did the lipid accumulation phase start? The latter is important since it is known that the fatty acid profile for thraustochytrids are very dependent on their growth phase, including their preferentially using SFA for energy when starving and their phospholipids containing more PUFA then their TAG-fraction do. Hence, any difference in growth between the cultures might also have an impact on their fatty acid profile.
  4. Since this is about improving bioprocesses: providing the numbers for the different fatty acids as both g/L (and % of total lipid) is needed. Of course, when the total lipid is provided both as g/L and g/biomass, which you should do, it is possible to calculate the others, but as a reader I prefer the authors to analyze their data providing me with numbers that can be compared directly.

A: According to comments 2 and 3, table 2 and more figures were improved as you suggested.

In this work we focus on exploring the interaction between the protist and the bacteria, and to show how such interaction causes changes in the fatty acid profiles. The results found so far would lead us to carry out new factorial tests for a better understanding of the current findings, with particular attention to the role of pH.

  1. The conclusion part should be rewritten

A: The conclusion was rewritten

Minor points

  1. Abstract, L1: “preferably aquatic” – do you know of any non-aquatic thraustochytrid? “marine” might have been a correct word. A: L1 corrected
  2. Abstract L 2: last words are just not in the right order A: L2 corrected
  3. Abstract L4 Why is “of fish” added? A: L4 corrected
  4. Abstract L5: you did not test cocultures of protists in general, only one strain. Rewrite this and the following lines A: L5 and following lines corrected
  5. Abstract last line: “technology with high nutritional attributes”? I can guess what you mean, but you need to rewrite. A: rewritten

I have not gone as detailed through the sentences later in the manuscript, I suggest too much rewriting

  1. P2 Thraustochytrids bioaccumulate: Not the best word. Fish may bioaccumulate, thraustochytrids produce. A: P2 corrected
  2. P3 and throughout: the names of microorganisms are only written fully the first time, then the genus name is to be abbreviated to the initial. A: suggestion implemented
  3. 2.1: I would like to know the origin of the 6 you found – from which of your three sources? And why did you only sequence two of them. That might make sense if you picked the best two auxin producers. In that case, this information needs to be put after 2.2. A: now all details are specified
  4. 2.2 Line 4, should it be 5.8 and not 5.6? A: L4 corrected
  5. 2.3 Why is that particular medium essential, there are many different media being used for this genus and even this strain? Indeed, you should probably explain why you chose this medium with a fairly low content of sugar and a relative high amount of nitrogen. A: It was the medium suggested for the maintenance of the strain according to ATCC collection.
  6. I suggest that you reorganize the 2.1-2.3 to make the story smoother. A: Thanks for your kind suggestion
  7. 2.4: another headline, this is not the methods section. And you can write it shorter: after 96 h, one treatment had a significantly higher biomass, and that was the only treatment that did not display a significant lower biomass after 144 h. Moreover, given that we are talking lipid accumulating organisms, would the increase between 96 and 144 h be lipids alone, or the total biomass? A: improved
  8. 2.5: The Table shows two strain, only one is commented on in the first paragraph. A: Improved
  9. The PCA analysis does not add anything to my understanding of the data. It would have made sense as a bridge to further growth experiments, e.g. if pH is that important, should you have tested growth of the thraustochytrid alone, but in a medium buffered to 6.2? A: improved
  10. Fig2: Grey and white is 96 and 144 h? and there are a couple of “y” in stead of “and” A: corrected
  11. Discussion. L2 “to project its use” I do not understand the phrase in this context.  And feed supplements or a feed supplement A: improved
  12. P 9: Our results are due to the fact …, I do not understand this – and I do not think this paragraph comes with necessary information not already covered well in the introduction. And since your Table 3 shows that auxin is not the most important factor – is it relevant at all. A: improved
  13. The paragraph starting with “different bacteria” is a justification of the study. If it were to be in the Paper, it would fit better in the introduction A: changed as you suggest
  14. P9 when discussing the biomass, please consider what would be the aim of your process. Biomass can be both lipids and the proteins (lipid free biomass). If lipids is the intended product, the amount of lipids is the interesting number. A: improved
  15. P10 top, is it possible that lactic acid bacteria could lower the pH? A: yes, it is possible. Specified in the manuscript
  16. Material and methods seems to be sufficient, perhaps a bit more detailed than necessary.  P12 3 lines above 4.7: The identification of fatty acids will be: was? A: corrected
  17. Supplementary Table 1 was not well shon in the manuscript – some of the right hand columns are missing A: corrected

Reviewer 2 Report

1- Application of marine biotechnology is a big field; I recommend to cite some recent reviews which cover this research field. For example,

Immobilization of Penaeus merguiensis alkaline phosphatase on gold nanorods for heavy metal detection (https://www.sciencedirect.com/science/article/pii/S0147651316304274)

Improved features of a highly stable protease from Penaeus vannamei by immobilization on glutaraldehyde activated graphene oxide nanosheets (https://www.sciencedirect.com/science/article/abs/pii/S0141813019305616)

 Production of newfound alkaline phosphatases from marine organisms with potential functions and industrial applications (https://www.sciencedirect.com/science/article/abs/pii/S1359511317310590)

Marine chitinolytic enzymes, a biotechnological treasure hidden in the ocean? (https://link.springer.com/article/10.1007/s00253-018-9385-7)

2. Overall, adequate Data/information have been reported/presented. However, there needs scientific explanation/reasoning (e.g. why/how) of the Data which mostly lacks.

3. Besides, the Results and Discussion (sub-titles) should correspond with the Materials and Methods (sub-titles). The authors could look into this and revise/amend accordingly.   

4Conclusions: Should summarize the significant findings of ALL THE MAJOR STUDIES, and thus needs to be extended further within the scopes. 

Author Response

A: Dear Reviewer, we are very grateful for all your comments, which helped us improve the quality of our work.

Comments and Suggestions for Authors

1- Application of marine biotechnology is a big field; I recommend to cite some recent reviews which cover this research field. For example,

Immobilization of Penaeus merguiensis alkaline phosphatase on gold nanorods for heavy metal detection (https://www.sciencedirect.com/science/article/pii/S0147651316304274)

Improved features of a highly stable protease from Penaeus vannamei by immobilization on glutaraldehyde activated graphene oxide nanosheets (https://www.sciencedirect.com/science/article/abs/pii/S0141813019305616)

 Production of newfound alkaline phosphatases from marine organisms with potential functions and industrial applications (https://www.sciencedirect.com/science/article/abs/pii/S1359511317310590)

Marine chitinolytic enzymes, a biotechnological treasure hidden in the ocean? (https://link.springer.com/article/10.1007/s00253-018-9385-7)

A: thanks for your suggestion, now improved

  1. Overall, adequate Data/information have been reported/presented. However, there needs scientific explanation/reasoning (e.g. why/how) of the Data which mostly lacks.

A: improved in the new version of the manuscript

  1. Besides, the Results and Discussion (sub-titles) should correspond with the Materials and Methods (sub-titles). The authors could look into this and revise/amend accordingly.   

A: Corrected

4Conclusions: Should summarize the significant findings of ALL THE MAJOR STUDIES, and thus needs to be extended further within the scopes

A: Rewritten
